# Metalloproteomics Reveals Multi-Level Stress Response in *Escherichia coli* When Exposed to Arsenite

**DOI:** 10.3390/ijms25179528

**Published:** 2024-09-02

**Authors:** James Larson, Brett Sather, Lu Wang, Jade Westrum, Monika Tokmina-Lukaszewska, Jordan Pauley, Valérie Copié, Timothy R. McDermott, Brian Bothner

**Affiliations:** 1Department of Chemistry and Biochemistry, Montana State University, Bozeman, MT 59717, USA; 2Department of Microbiology and Cell Biology, Montana State University, Bozeman, MT 59717, USA; 3Department of Land Resources and Environmental Sciences, Montana State University, Bozeman, MT 59717, USA

**Keywords:** arsenic, proteomics, toxicity, bacteria, metal

## Abstract

The *arsRBC* operon encodes a three-protein arsenic resistance system. ArsR regulates the transcription of the operon, while ArsB and ArsC are involved in exporting trivalent arsenic and reducing pentavalent arsenic, respectively. Previous research into *Agrobacterium tumefaciens* 5A has demonstrated that ArsR has regulatory control over a wide range of metal-related proteins and metabolic pathways. We hypothesized that ArsR has broad regulatory control in other Gram-negative bacteria and set out to test this. Here, we use differential proteomics to investigate changes caused by the presence of the *arsR* gene in human microbiome-relevant *Escherichia coli* during arsenite (As^III^) exposure. We show that ArsR has broad-ranging impacts such as the expression of TCA cycle enzymes during As^III^ stress. Additionally, we found that the Isc [Fe-S] cluster and molybdenum cofactor assembly proteins are upregulated regardless of the presence of ArsR under these same conditions. An important finding from this differential proteomics analysis was the identification of response mechanisms that were strain-, ArsR-, and arsenic-specific, providing new clarity to this complex regulon. Given the widespread occurrence of the *arsRBC* operon, these findings should have broad applicability across microbial genera, including sensitive environments such as the human gastrointestinal tract.

## 1. Introduction

Arsenic is a carcinogenic toxin that is ubiquitous in the environment [1]. Environmental arsenic is most commonly found in one of two inorganic forms: trivalent (arsenite, As^III^) and pentavalent (arsenate, As^v^) species. While both As^III^, and As^V^ are toxic to organisms, As^III^ is generally the more toxic form [2,3]. Despite the known toxicity, millions of people across the world consume drinking water containing arsenic levels above the WHO-recommended concentration of 5 ppb, emphasizing that arsenic remains a global health concern [4].

Due to its pervasiveness in the environment, organisms have developed resistance and detoxification mechanisms against As^III^ and As^V^ [5,6,7,8]. The proteins responsible for this in prokaryotes are encoded by genes arranged on *ars* operons [9]. In *Escherichia coli*, it is a single *arsRBC* operon that encodes for ArsR, a transcriptional repressor, ArsB, an arsenite efflux pump, and ArsC, an arsenate reductase [8,10]. As^III^ competitively binds to ArsR, causing a conformational change that results in ArsR releasing from the *ars* promoter, and increasing transcription of the operon. ArsC reduces As^V^ to the more toxic As^III^. This counterintuitive transformation is believed to be advantageous because As^III^ can be actively exported from the cell by ArsB. *E. coli* strains harboring the *arsRBC* operon have increased resistance to arsenic toxicity [11].

Recent reports have shown that ArsR is involved in regulation beyond that of the *ars* operon. *Agrobacterium tumefaciens* 5A has an intricate relationship with arsenic. This organism has an arsenite oxidation system that can produce energy under low phosphate conditions [12,13]. In addition, it has two arsenic detoxification loci that contain *arsRBC*-like genes encoding for a total of four ArsR proteins (ArsR1-4) [12]. Differential transcriptomics of individual *arsR* knockouts have shown that these ArsR proteins have wide-scale regulatory influences across all physiological processes [14]. Furthermore, ArsR proteins displayed a regulatory hierarchy of each other’s transcription. All of these *arsR* genes have been shown to regulate metal homeostasis proteins.

The seminal findings in *A. tumefaciens* may have been unique to arsenic-oxidizing organisms with sophisticated ArsR regulatory networks. We have previously investigated the effects that the presence/absence of the *arsRBC* operon has on the global native metalloproteome in *E. coli* strains under As^III^- and As^V^-stressing conditions [15]. In this study, we used size-exclusion chromatography coupled with an inductively coupled mass spectrometer (SEC-ICPMS) to analyze the metalloproteins (proteins that bind metal) in their native state. Changes to the magnesium, iron, zinc, and nickel metalloproteome were detected upon arsenic treatment. The changes differed depending on the presence of the *arsRBC* operon. Furthermore, protein cofactors and metal uptake were influenced by the *arsRBC* operon under arsenic stress. Our results show that the presence of the *arsR* gene may alter metalloproteins in *E. coli* but may be caused by the presence of *arsB* or *arsC*. Additionally, SEC-ICPMS does not identify changes to individual protein expression levels but provides a global view of the distribution of metal proteins and protein complexes.

Here, we used a differential proteomics approach to identify specific metalloproteins affected by the presence of ArsR under high As^III^ stress using *E. coli* containing *arsRBC*, *arsR*, or a deletion mutant lacking the *arsRBC* locus. We elucidate a common As^III^ stress response in all strains regardless of the presence of the *arsR.* Additionally, we identify proteins whose expression is affected by the presence of ArsR under high As^III^ stress. This work provides new information about biological stress responses to As^III^ and the regulatory effects of ArsR.

## 2. Results

### 2.1. arsR Confers Arsenic Resistance

To investigate the changes to the metalloproteome under a sublethal yet significant degree of As^III^ stress, AW3110 and the *arsR*-complement were exposed to 100 µM As^III^. Growth of treated cells was reduced compared to the control cells, but cells continued growing (Figure 1a). Interestingly, As^III^ had less of an effect on the *arsR*-complement than AW3110, despite lacking the As^III^ efflux pump gene, *arsB*. As expected, the wild-type strain K-12 required elevated concentrations of As^III^ to achieve a similar reduction in growth (Figure 1b). Based upon the levels of As^III^ required to induce significant decreases in growth rate across strains, we designated the K-12 samples given 1 mM As^III^, and the AW3110 and *arsR*-complement samples treated with 100 µM As^III^ as the “high As^III^ samples/groups.” We designated the control samples and the K-12 samples that received the 100 µM As^III^ treatment as the “low/no As^III^ stress samples/groups”.

### 2.2. Changes to the Metalloproteome during As^III^ Stress

We speculated that protein expression would be altered due to As^III^ stress and by the presence of ArsR. To investigate this, we employed shotgun proteomics to determine the relative abundance of proteins expressed by each strain under the different growth conditions. We identified 1840 proteins from the 4328 predicted protein-coding genes in K-12 MG1655 and 4243 predicted protein-coding genes in the AW3110 substrain (Appendix A) [16,17]. We choose to focus our analysis on metalloproteins since the *arsRBC* operon has been shown to differentially affect the global native metalloproteome under As^III^ stress [15]. To do this, we filtered the identified proteins using the DAVID functional analysis (see methods) (Appendix A). From the identified protein pool, 619 (33.6%) were categorized as metalloproteins. Principal component analysis based on relative abundance revealed that AW3110 and *arsR* control samples grouped closely together and the As^III^ treated samples grouped close together (Figure 2a). The K-12 control and K-12 low As^III^ stress grouped close together, while the high As^III^ stress samples were well separated from the other K-12 groups along both axes. Replicates in every treatment were tightly clustered with each other. A one-way ANOVA with a false discover rate cutoff of 0.001 revealed 363 metalloproteins significantly changed between all groups (Appendix A). These data suggest that large changes in the metalloproteome occurred between sample groups that we interpret as deriving from As^III^ stress and strain genotypes.

To better understand the differences in our proteomics data, we employed a hierarchical clustering analysis (HCA) on the top 30 dysregulated proteins. The results from our PCA were mirrored when the data were assessed in this fashion (Figure 2b). The HCA clustered K-12 samples apart from the AW3110 and *arsR*-complement samples. Within those clusters, high As^III^ stress samples clustered with each other, and the low/no As^III^ stress samples clustered together. Clustering based on changes in protein abundance highlighted strain differences, As^III^ stress, and the presence of ArsR (Figure 2b). As expected, ArsR (Uniprot accession number P37309) was present in *arsR*-complement and arsenic-treated K-12 samples, while ArsC was present only in As^III^-treated K-12 samples. Within the strain difference protein cluster, several galactitol metabolism-related proteins were identified. In AW3110 and the *arsR*-complement, galactitol-1-phosphate dehydrogenase, GatD (P0A9S3), and tagatose-1,6-bisphosphate aldolase, GatY (P0C8J6) were more abundant than in K-12. High As^III^ stress stimulated the expression of several metal-cofactor biosynthesis proteins. These were molybdenum cofactor biosynthesis protein B (MoaB, P0AEZ9), cysteine desulfurase (IscS, P0A6B7), and [Fe-S] cluster assembly scaffold protein (IscU, P0ACD4) (Figure 3). Six proteins seemed to be influenced by the presence of ArsR including proteins that were differentially expressed between AW3110 and the *arsR* high As^III^ samples (Figure 2b). Several of these proteins were part of the TCA cycle. These proteins were acetyl-coenzyme A synthetase (P27550), isocitrate dehydrogenase (P08200), fumarase A (P0AC33) and fumarase B (P14407). These results suggest that *E. coli* responds to As^III^ by increasing metallocofactor biosynthesis proteins and that TCA cycle proteins appear to be affected by the presence of ArsR.

Having seen that molybdenum cofactor B, IscS and IscU were affected by the As^III^ treatment, we searched our data for proteins involved in these cofactor synthesis pathways. All of the Isc [Fe-S] cluster assembly proteins, including the chaperone and accessory proteins, HscA, HscB, and ferredoxin, were present and increased in intensity under high As^III^ stress (Appendix A). We did not detect IscX in any of our samples. In all groups, the Isc proteins increased in abundance upon high As^III^ stress. We also identified all of the Suf [Fe-S] cluster assembly proteins (Appendix A). Out of the six Suf proteins, only SufA and SufS increased upon As^III^ stress (Appendix A). The others decreased or did not change in abundance. All of the molybdenum cofactor assembly proteins, except for MoaA, increased in abundance in the high As^II^ stress groups (Appendix A). These data show that all Isc [Fe-S] cluster synthesis proteins, two Suf [Fe-S] synthesis proteins, and the majority of the molybdenum cofactor synthesis proteins are more highly expressed when *E. coli* is under high As^III^ stress.

To decipher the effects caused by As^III^ on TCA protein expression, as well as the role of ArsR, we examined the change in expression of the TCA associated metalloenzymes between the control and high As^III^ stress groups (Appendix A). For every enzymatic step of the TCA cycle containing a metalloprotein, the change in expression between control and high As^III^ stress was different for AW3110 and the *arsR*-complement. This trend was also observed when the statistical analysis included non-metal-containing TCA cycle enzymes (Figure 4). Therefore, ArsR impacts the abundance of TCA enzymes under high As^III^ stress.

### 2.3. Intracellular Metal Changes

Based on the previous literature [18], we hypothesized that the intracellular metal concentrations would be different upon arsenic treatment, and that the presence of *arsRBC* operon and ArsR would impact observed changes, challenging us to quantify intracellular metal concentrations (cytosolic metal and protein-bound metal). To investigate this, we used ICP-MS, which allowed us to quantify many metals simultaneously. The metals detected by SEC-ICPMS are discussed below, along with noteworthy results. Additional metal concentrations can be found in Appendix A. As expected, arsenic concentrations inside the cells increased with As^III^ stress levels (Figure 5). Interestingly, the *arsR*-complement strain had a higher concentration of arsenic than the K-12 and AW3110. The low As^III^ stress K-12 samples and the high As^III^ stress AW3110 samples resulted in the same intracellular arsenic concentrations and high As^III^ stressed *arsR*-complement resulted in the highest concentration across sample groups. In all strains, high As^III^ stress led to a significant increase (*p* < 0.05) in Ca, Na, Mo, Pb, and Mn, and a significant decrease in Fe, and Mo. Copper was unaffected by the presence of As^III^. Magnesium concentrations decreased under high As^III^ stress only in K-12 with no change in AW3110 or the *arsR*-complement. Similarly, zinc only changed in the *arsR*-complement, where it increased with higher levels of As^III^. Additionally, copper, manganese, and calcium concentrations were significantly different (*p* < 0.05) between AW3110 and the *arsR*-complement controls. Sodium and iron concentrations were also different but to a lesser extent (*p*-value of 0.05, and 0.08, respectively). These results show that high As^III^ stress impacts intracellular metal concentrations and that the presence of the *arsR* gene plays a role in metal homeostasis both in the presence and absence of As^III^.

## 3. Discussion

The goal of this work was two-fold: (1) identify metalloproteins affected by As^III^ and (2) characterize ArsR regulatory influences that extend beyond the *ars* operon. The near-isogenic *E. coli* strains [19,20] used in this study differed primarily based on the presence or absence of specific *ars* genes and their encoded As^III^ resistance functions. K-12 has the complete *arsRBC* operon and thus should display maximal As^III^ resistance, whereas AW3110 lacks the *arsRBC* operon and thus is more sensitive. The use of the AW3110 carrying the *arsR* gene under control of the native promoter allowed us to selectively examine the function of this transcription factor, uncoupled from *arsB* and *arsC*, to assess if its regulatory bandwidth extends beyond the *ars* operon as we recently demonstrated with *Agrobacterium tumefaciens* [14].

The differential proteomics workflow showed these *E. coli* cells clustered based on their *arsRBC* and *arsR* differences (Figure 2), indicating that the presence and absence of the *arsRBC* genes were quite important with regard to the cellular metalloproteomics profiles in response to As^III^ stress. However, all three strains had a common response to As^III^ stress that involved the activation of two metallocofactor biosynthetic pathways (Figure 3 and Appendix A). This included proteins encoded by the *moaABCDE* gene cluster, although the upregulation of MoaA in response to As^III^ was only observed in the *arsR*-complement strain. Moa proteins are involved in the biosynthesis of the molybdenum cofactor (Moco), a molybdopterin found in molybdeoenzymes [21]. Molybdoenzymes catalyze versatile and essential redox reactions in carbon nitrogen, and sulfur cycles [22]. Similarly [Fe-S] clusters catalyze a wide range of redox reactions in virtually every cellular process [23]. Even the Moco synthesis pathway requires a [4Fe-4S] cluster to catalyze the initial step in Moco synthesis [24]. The Isc [Fe-S] cluster synthesis proteins are all upregulated upon high As^III^ stress (Figure 3 and Appendix A), but only SufA and SufS are upregulated in the Suf [Fe-S] cluster synthesis pathway (Appendix A). In *E. coli,* the Isc pathway is regarded as the housekeeping [Fe-S] pathway for cluster synthesis, whereas the Suf pathway for [Fe-S] cluster synthesis responds to oxidative stress [24,25,26,27]. Since As^III^ is a well-known inducer of oxidative stress [28,29,30,31], these results were not expected. As^III^ has a high affinity for thiols and reacts with reduced cysteines in proteins, which can alter protein structure and impair catalysis [32,33,34], both of which contribute to As^III^ toxicity [32]. IscR, the first gene product of the *isc* operon, is a potential target for As^III^ interactions, as it possesses three cysteine residues critical for Fe-s cluster ligation. IscR regulates over 40 genes and is a transcriptional regulator that plays both an activating as well as a repressing role depending on its cofactor status [35]. In its holo [2Fe-2S]-containing form, IscR represses *isc* transcription, while the apo form represses Suf expression [35,36]. As^III^ may disrupt [2Fe-2S] cluster incorporation, leading to the expression of the Isc proteins. Our data support this, since the increased abundance of IscR did not correlate to a decreased abundance of other Isc proteins (Appendix A). In addition to IscR, the *suf* operon is under the regulatory control of two other transcription factors harboring potential As^III^-reactive thiols. The ferric uptake regulator (Fur) represses Suf exclusively when it is coordinating iron [37]. Similarly, the transcription factor OxyR stimulates Suf expression, but only when the cysteines of OxyR are in a certain disulfide bridge configuration [38,39]. Isc and Suf expression is under complex regulatory control and is likely affected by the disruption of protein function through high-affinity As^III^–cysteine interactions.

Recently, it has been demonstrated that ArsR has widespread effects that extend beyond the known *arsR*-regulated operon and arsenic resistance [14,15,40]. To isolate changes specific to the presence of ArsR in *E. coli*, we compared AW3110 with AW3110 complemented with *arsR*, finding that ArsR influences metal homeostasis, and thus corroborating its role in *A. tumefaciens* wherein the ArsRs control metal homeostasis, particularly related to iron, copper, and nickel [14]. The disruption of metal homeostasis likely plays a role in arsenic toxicity and the preservation of that homeostasis is expected to provide a fitness advantage. Importantly, complementing *arsR* in AW3110 enhanced the growth rate in the presence of As^III^ (Figure 1) despite significantly greater As bioaccumulation in the *arsR*-complemented AW3110 (Figure 5). We attribute this to ArsR acting to effectively absorb As^III^ and thereby reduce free As^III^ in the cell that would otherwise damage numerous enzymes and growth. This phenomenon has also previously been suggested [41]. 

In the presence of ArsR, the *E. coli* metalloproteome displayed a response to high As^III^ stress (Figure 2b), exhibiting conferred As^III^ resistance (Figure 1), which was associated with TCA cycle protein expression. At least one protein in every enzymatic step of the TCA cycle had altered expression due to the presence of ArsR (Figure 4). As^III^ exposure has been linked to the inhibition of α-ketoglutarate dehydrogenase enzymes in *Agrobacterium tumefaciens* 5A, shunting metabolites away from the TCA cycle [42]. When ArsR was present, the subunits of α-ketoglutarate dehydrogenase increased in expression under high As^III^ stress, which can be attributed to maintaining TCA cycle function. Conservation of energy, however, is a common stress response in *E. coli* [43]. Yet As^III^ exposure to cells containing *arsR* appears to upregulate the TCA cycle, which can reasonably be inferred as increasing electron flow to electron transport activity and thus increasing cellular energy. This phenomenon can be attributed to increasing the abundance of TCA cycle intermediates, which has been proposed to play an active role in the chelation of toxic metals under metal-induced stress [44]. Indeed, several of these TCA metabolites have been shown to aid in alleviating toxicity caused by zinc, copper, and aluminum. Citrate, malate, and succinate have all been shown to neutralize these toxic metals through chelation. Additionally, aluminum and zinc toxicity has been demonstrated to diminishwith increased production of oxaloacetate derivatives. TCA enzymes may be upregulated in the presence of ArsR to provide metabolites that can chelate As^III^. ArsR influences the expression of the majority of TCA-cycle enzymes and is presumably linked to the increased As^III^ resistance in the *arsR*-complement.

## 4. Materials and Methods

### 4.1. Construction, Cloning, and Transformation of the arsR Plasmid

The complete *arsR* coding region along with its native promoter from K-12 wild-type strain W3110 was PCR-cloned with primers ArsR-F (AAATTAATTAATATTACCTTCCTCTGCACTTAC) and ArsR-R (AAACCTAGGTTAACTGCAAATGTTCTTACTGT) using a pCR2.1 TOPO TA Cloning kit (Invitrogen™, Waltham, MA, USA). The resulting pCR2.1-p-*arsR* plasmid was transformed into *E. coli* AW3110 following the cloning kit manual instructions. Successful transformation was confirmed using GENEWIZ’s Sanger sequencing service (Azenta Life Sciences, Burlington, MA, USA). We refer to this complemented strain as AW3110 (p-*arsR*) to indicate the presence of the *arsR* gene under the control of its native promoter carried on the pCR2.1 plasmid.

### 4.2. Culturing Conditions

All *E. coli* cultures were grown in 200 mL of LB broth as batch cultures at 37 °C. LB broth was prepared in-house using tryptone (MilliporeSigma, Burlington, MA, USA), yeast extract (Thermo Scientific, Waltham, MA, USA) and sodium chloride (Fisher Scientific, Waltham, MA, USA) [45]. The K-12 MG1655 (wild type) and AW3110 (chloramphenicol cassette marked Δ*arsRBC* mutant) cell lines were a gift from Dr. Barry Rosen and are described in the original publication [11]. Cultures were inoculated with 2 mL of an overnight culture and tested under seven conditions (*n* = 3). To elicit a similar stress level in K-12 and AW3110 and based upon previous research [15], we chose to use 1 mM As^III^ for K-12 and 100 µM As^III^ for AW3110 and the *arsR*-complement. In addition, K-12 was also given 100 µM As^III^ as a control for As^III^ concentrations between K-12 and AW3110 strains. As^III^ was administered 2 h post-inoculation, and growth was recorded via OD_600_ measurements every hour. Two hours after the addition of As^III^, 50 mL of each culture was removed. Cells were pelleted by centrifugation (4000× *g*, 10 min, 4 °C) and stored at −80 °C until further analyses. AW3110 and the *arsR*-complement were grown under chloramphenicol and kanamycin selection, respectively.

### 4.3. Protein Extraction

Cell pellets were washed three times with 5 mL of lysis solution (200 mM ammonium acetate, pH 7) by gentle shaking. After each wash, cells were pelleted by centrifugation (1500× *g*, 5 min, 4 °C). Cells were then resuspended in 800 µL for lysis using Matrix E (MPBio, Irvine, CA, USA) on a FastPrep-24 5G bead beater (MPbio) at a speed of 6.0 m/s for 30 s. Cell debris and matrix were pelleted at 18,000× *g* for 20 min at 4 °C, and the soluble fractions were collected for further analyses. Protein concentrations were determined using a Bradford assay [46].

### 4.4. Shotgun Proteomics

Protein samples were concentration-matched, reduced, alkylated, and cleaned by chloroform/methanol extraction prior to digestion with sequencing grade modified porcine trypsin (Promega, Madison, WI, USA) at the IDeA National Resource for Quantitative Proteomics at the University of Arkansas Little Rock. Tryptic peptides were then separated by reverse phase XSelect CSH C18 2.5 um resin (Waters, Milford, MA, USA) on an in-line 150 × 0.075 mm column using an UltiMate 3000 RSLCnano system (Thermo, Waltham, MA, USA). Peptides were eluted using a 60 min gradient from 98:2 to 65:35 solution A:B ratio. Eluted peptides were ionized by electrospray (2.4 kV) followed by mass spectrometric analysis on an Orbitrap Eclipse Tribrid mass spectrometer (Thermo). MS data were acquired using the FTMS analyzer in profile mode at a resolution of 120,000 over a range of 375 to 1200 *m*/*z*. Following HCD activation, MS/MS data were acquired using the ion trap analyzer in centroid mode and normal mass range with a normalized collision energy of 30%. Peptides fragmented during the acquisition were used for the basis of identification. Proteins were identified by database search using MaxQuant (Max Planck Institute, Munich, Germany) with a parent ion tolerance of 3 ppm and a fragment ion tolerance of 0.5 Da. Scaffold Q+S (Proteome Software, version 5.0.1) was used to verify MS/MS-based peptide and protein identifications. Protein identifications were accepted if they could be established with less than 1.0% false discovery and contained at least 2 identified peptides. Protein probabilities were assigned by the Protein Prophet algorithm [47].

### 4.5. Curation of Metalloproteome from Identified Proteins

To generate the metal-containing protein list, the identified proteins were submitted to the online functional annotation tool on the Database for Annotation, Visualization and Integrated Discovery (DAVID) [48,49]. The results from DAVID were manually searched against metal keywords to determine if a protein was a metalloprotein. Search terms were “zinc”, “zn”, “iron”, “4fe”, “ferredoxin”, “ferrous”, “ferric”, “siderophore”, “heme”, “2fe”, “nickel”, “magnesium”, “mg”, “cobalt”, “cobalamin”, “arsenic”, “arsen”, “ars”, “sodium”, “na+”, “na2+”, “manganese”, “mn”, “copper”, “cupric”, “molybdenum”, “molybdopterin”, “potassium”, “calcium”, and “metal”.

### 4.6. Statistical Analysis of the Metalloproteome

The identified metalloproteins were input into the Metaboanalyst online software suite (version 6.0 https://www.metaboanalyst.ca/) [50]. Upon input, data were normalized by sum, mean-centered, divided by the standard deviation of each variable for scaling, and log_10_-transformed. A one-way ANOVA with an FDR of 0.001 and a post hoc implemented Tukey’s HSD with a threshold value of 0.001 was used to identify significantly different proteins of interest based upon comparisons between normalized, relative protein abundances. Individual box plots of normalized proteins of interest were generated from the ANOVA analysis. Principal component analyses were performed using all sample groups and on only the sample groups that exhibited high As^III^ stress. Hierarchical clustering analysis was performed using Euclidian distances with a Ward clustering method of an applied ANOVA analysis to generate top 30 feature heatmaps. The statistical analysis was repeated using non-metal TCA cycle enzymes.

### 4.7. Total Metal Analysis of Soluble Fractions 

Proteomes collected 2 h post arsenic dosing (soluble lysate) were diluted to a protein concentration of 4 mg/mL. A total of 800 µg of protein from each sample was digested in 20% *wt*/*v* optima grade HNO_3_ (Fisher, Waltham, MA, USA) at 99 °C for 30 min. Precipitate was pelleted by centrifugation and 150 µL of the supernatant was diluted to 3 mL for metal analysis with 2% HNO3, 0.5% HCl solution. Samples were analyzed on an Agilent 7800 ICP-MS with an Agilent SPS4 autosampler. Metal concentrations were determined using standard curves generated from serial dilution of a commercially available environmental calibration standard (CPI International, Santa Rosa, CA, USA). An internal standard mix (Agilent, Santa Clara, CA, USA) was added to the samples using a T-junction immediately before the nebulizer. The ICP-MS parameters were auto-tuned using an ICP-MS tuning solution (Agilent).

## 5. Conclusions

In all three strains, the molybdenum cofactor and Isc [Fe-S] cluster assembly increased in expression under high As^III^ stress indicating a conserved response to As^III^ stress. During oxidative stress, [Fe-S] cluster synthesis is expected to be mediated by Suf proteins, not Isc. We suspect that As^III^ may be impairing regulatory proteins causing Isc expression. In addition, we show a stress response unique to strains containing the *arsR* gene. The presence of ArsR has a regulatory role on the expression of TCA cycle proteins under high As^III^ stress. ArsR also conferred As^III^ resistance compared to the AW3110 parent strain. This may be connected to the expression of TCA cycle enzymes but remains only hypothetical.

This work provides important evidence that expands upon hypotheses regarding the bacterial arsenic stress response and the role(s) of ArsR as a transcriptional regulator. Prior RNASeq-based work on this topic concerned the soil bacterium *A. tumefaciens* [14], whereas the current effort focused on *E. coli*, which is a common member of the gut microbiome. As such, this has significant relevance to understanding the full impact of arsenic exposure and toxicity to roughly 200 million humans worldwide consuming arsenic contaminated drinking water [51,52]. The human gastrointestinal tract microbiome constitutes the first response to ingested arsenic, which is governed to a large degree by ArsR [53]. Consequently, fully appreciating the regulatory bandwidth of this arsenic-sensitive regulator is critical towards appreciating how and why the human gut microbiome responds to arsenic exposure, and how that can influence host health and welfare. 

## Figures and Tables

**Figure 1 ijms-25-09528-f001:**
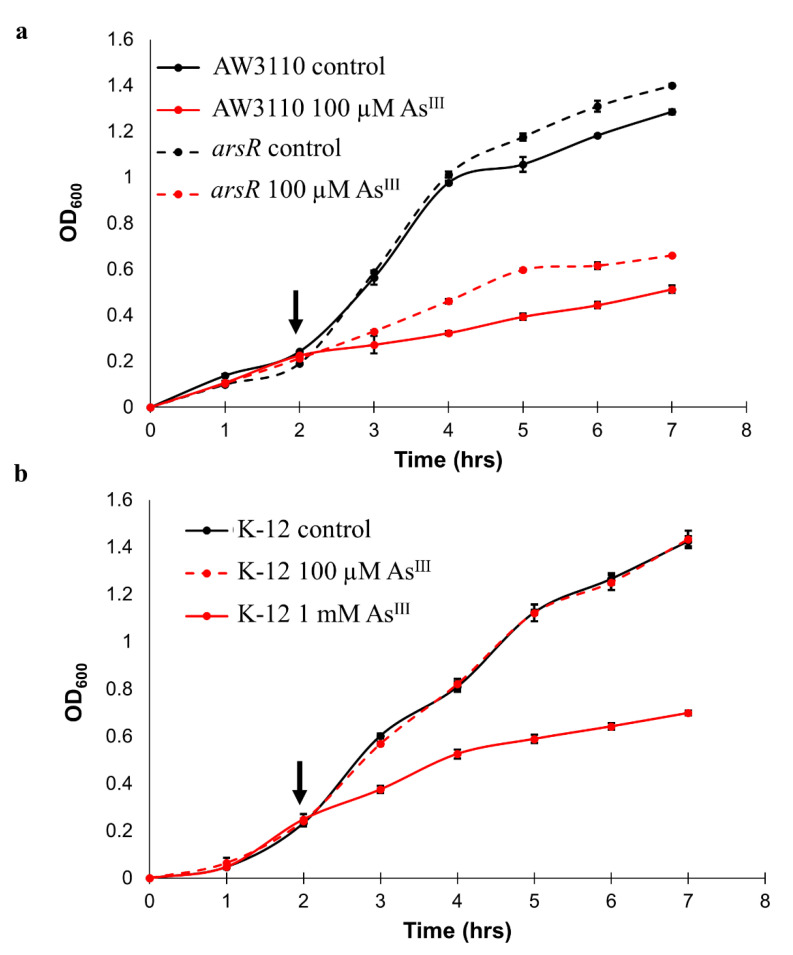
ArsR confers As^III^ resistance. (**a**) Growth of AW3110 and *arsR*-complemented AW3110. Cultures were grown in triplicate (*n* = 3) in the presence (red) and absence of arsenic (black) for AW3110 (solid line) and *arsR*-complement (dashed). A 100 µM As^III^ treatment was given at 2 h post-inoculation. (**b**) Growth of K-12. Cultures were grown in triplicate (*n* = 3) in the presence (red) and absence of arsenic (black). The arsenic concentrations were 100 µM (dashed) or 1 mM As^III^ (solid). Arsenite was added 2 h post-inoculation, indicated by the arrow. Samples were collected 2 h later for proteomics analysis. Error bars are ± 1 standard deviation.

**Figure 2 ijms-25-09528-f002:**
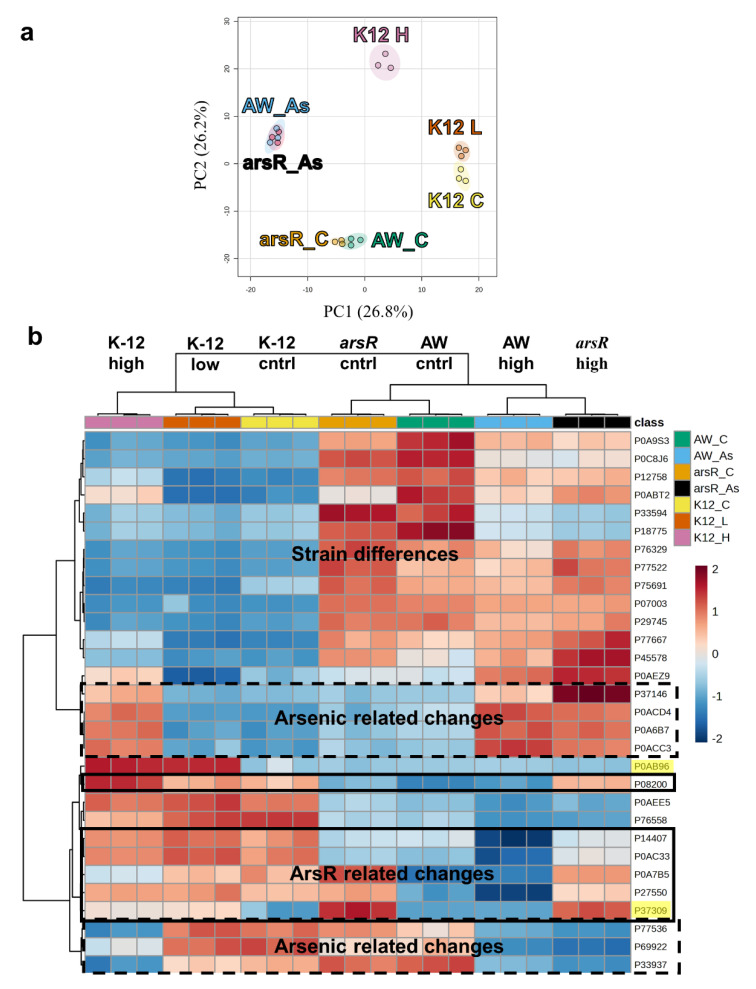
Statistical analysis of the metal-related proteome. (**a**) PCA analysis of *E. coli* strains K-12, AW3110, and the *arsR*−complement with and without As^III^ stress. The labels are as follows: K12 H is K-12 with high As^III^ stress, K12 L is K-12 with low As^III^ stress, K12 C is the unstressed K-12 control, AW_As is the AW3110 high As^II^ stress, AW_C is the AW3110 control, arsR_As is the *arsR*-complement high As^III^ stress, and arsR_C is the *arsR*-complement control. (**b**) Hierarchical clustering based on the top 30 features of a one-way ANOVA. Protein cluster based on strain differences, high arsenite stress, and the presence of ArsR. Proteins are given as their Uniprot accession numbers. Proteins whose expression was altered based upon the presence of ArsR are outlined with solid boxes, proteins clustered due to high As^III^ stress are outlined with dashed boxes, and clusters due to strain differences are not outlined. The ArsR (P37309) and ArsC (P0AB96) proteins and highlighted.

**Figure 3 ijms-25-09528-f003:**
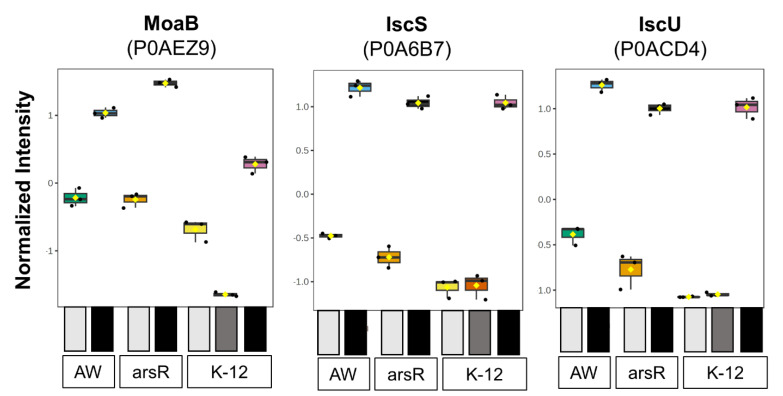
Metallocofactor proteins affected by high As^III^ stress. Based on a one-way ANOVA, these proteins increased in abundance in all strains under high As^III^ stress and were in the top 30 metalloproteins. Increasing darkness indicates the severity of As^III^ stress. The no As^III^ control for each strain is white, low As^III^ stress is grey, and high As^III^ stress is black. The boxes are colored to match the sample group colors in Figure 2.

**Figure 4 ijms-25-09528-f004:**
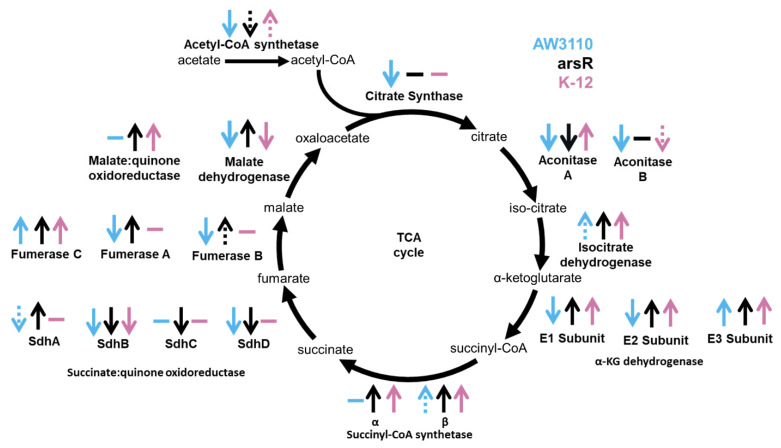
ArsR regulates expression of TCA cycle protein expression under As^III^ stress. The expression of TCA cycle proteins under high As^III^ stress compared to the no As^III^ controls is indicated by arrows and is color coordinated. AW3110 is indicated in blue, the *arsR*-complement is in black, and K-12 is in pink. An up arrow indicates an increase in expression. A down arrow indicates a decrease in expression. No arrow means the expression was unchanged. Dashed arrows indicate a marginal change in expression.

**Figure 5 ijms-25-09528-f005:**
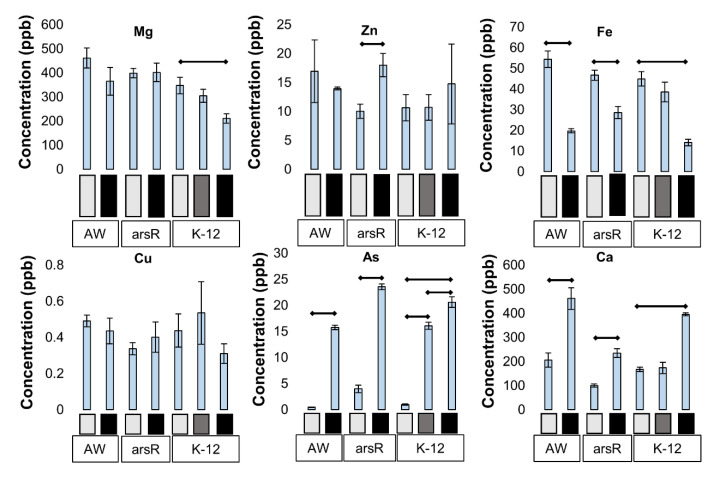
Soluble intracellular metal concentrations. The soluble cellular metal concentrations from each condition (*n* = 3) were measured with ICP-MS. Bars denote significant differences at *p*-value < 0.05. Only significance intragroup is displayed.

## Data Availability

Data is contained within the article or Appendix A.

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
