# Peer review of "Metalloproteomics Reveals Multi-Level Stress Response in Escherichia coli When Exposed to Arsenite"

_ijms, 2024, doi:10.3390/ijms25179528_

Round 1

Reviewer 1 Report

Comments and Suggestions for Authors

Dear authors

Here send the manuscript in format docx, and I added the line number and I correct someone word with slash.

In addition, here I write some comments:

Comments in the introduction section.

Line 41. “coun-sing” change by “counsing”

Line 43. “coun- terintuitive” change by “counterintuitive

Line45. Please, the word “toxcicity.” change by “toxicity”

Line 48. Please, to change “or- ganism ”by “organism

Line 49. Please, to change “detoxication” by “detoxification”

Line 59. Please to add the bibliographical citation

Line 53-56. Please to eliminste the white sheet.

Line 66. Please “de- pending” to change by “depending

Line 92. Please to change “publica- tion.” by “publication”.

Comments in the methods section.

Line 77. suggest change the title “” by “Clonning and transformation of construction arsR plasmid” .

Line 92. please to change “publica- tion” by “publication.”

Line 94. please to change “pre-vious” by “previous .”

Line 89. Please describe the characteriscs or kind LB broth there are differents types specify it and specify its brand.

Line 98. Please to change “ad- dition” by “addition”.

Line 103 . Please check the “200 m” to be ¿200 mM?.

Line 109. Please to add the bibliographical citation about bradford.

Line 120. Please correct this “spec- trometer” by “spectrometer”.

Lines 136-140. please check the word are writr correct, for example “4fe,”” or (“2fe”), are there correct?

Line 143.please to add the link web about Metaboanalyst.

Line 144. please correct “Hierarchal” by “Hierarchical”.

Line 151.please to change “dis- tances” by “distances”

Line 157-158. Please to change “via” by “by” and centrif- ugation correct it.

Line 178. Please to check “sam- ples”change by “samples”.

Line 214. Please to check this word in all text “Hierarchal to be “Hierarchical”.

Lines 184-209. Please check the next comments; in the caption of figure 1 check the word “cul-tures”, “pres-ence”, “condi-tions”, “rel- ative , “func-tional”, “Sam-ples”, “metallopro-teome”.

Lines 222-245.please to check it the “cluster- ing” and “clus-tered” , “sam- ples”, “differ-ence”, “biosyn- thesis ”, “pro- teins” .

Line 246. In the figure 3 caption to check “metallo- proteins”.

Figure 3. I have a question , the color of the box, What is it indicate?, incluide the meaning in the caption.

Lines 260. Please to check this words “Sup- plemental”,

Line 280. Please changed the title to Intercellular metal changes.

Lines281-297. Please to correct the words: Sim- ilarly, dif- ferent , respec- tively

In the figure 5 in the figure that showed the analyses of Zinc analyses, how do you explain the variation between assays specially in this analyses of the zinc, and the zinc quantity identical between k-12 high condition vs AW control condition?

Line 400. please to add E. coli in italics.

Good luck

Comments on the Quality of English Language

 Minor editing of English language required

Reviewer 2 Report

Comments and Suggestions for Authors

Comments to authors:

The manuscript entitled “Metalloproteomics Reveals Multi-Level Stress Response in Escherichia Coli When Exposed to Arsenite” by Larson et, al. described an extensive survey of arsR impact on AsIII induced proteome alteration. The study is well designed and clearly presented. I just have some issues (most of them are minor) for the authors to address, before this manuscript get suitable to be published.

1.     Lacking of integration between the results is a major concern. There are some disconnections and sharp turns in the logic flow of the results. For instance, why focus on TCA analysis after clustering? How does metal analysis connect to proteomic analysis in the scheme?

2.     For results, the purposes are usually missing. The authors dive directly into what has been done, instead of explaining why certain experiment needs to be carried out.

3.     In the TCA cycle analysis, changes are shown. But what is the exact impact, overall & target-specific?

4.     Why these metals are picked, in the metal analysis? What are the functional implications of each changed/unchanged result?

5.     The introduction part reads like a review, plus simplified methods section. More importantly, it failed to identify the gap of knowledge, i.e., what is known & what is unknown. Then, how may this work be able to fill the gap? Most importantly, how may filling such gap be important to be field?
